# Recreating the Trabecular Outflow Tissue on Implantable, Micropatterned, Ultrathin, Porous Polycaprolactone Scaffolds

**DOI:** 10.3390/bioengineering10060679

**Published:** 2023-06-02

**Authors:** Luke A. Beardslee, Justin R. Halman, Andrea M. Unser, Yubing Xie, John Danias, Magnus Bergkvist, Susan T. Sharfstein, Karen Y. Torrejon

**Affiliations:** 1Colleges of Nanoscale Science and Engineering, SUNY Polytechnic Institute, 257 Fuller Road, Albany, NY 12203, USA; 2Department of Ophthalmology, SUNY Downstate Health Sciences University, 450 Clackson Avenue, Brooklyn, NY 11203, USA; 3Glauconix Biosciences Inc., 251 Fuller Road, Albany, NY 12203, USA

**Keywords:** polycaprolactone, microfabrication, micropatterning, scaffold, tissue engineering, trabecular meshwork, myocilin, glaucoma

## Abstract

Glaucoma, where increased intraocular pressure (IOP) leads to damage to the optic nerve and loss of sight, is amongst the foremost causes of irreversible blindness worldwide. In primary open angle glaucoma, the increased IOP is a result of the malfunctioning human trabecular meshwork (HTM) cells’ inability to properly regulate the outflow of aqueous humor from the eye. A potential future treatment for glaucoma is to replace damaged HTM cells with a tissue-engineered substitute, thus restoring proper fluid outflow. Polycaprolactone (PCL) is a versatile, biodegradable, and implantable material that is widely used for cell culture and tissue engineering. In this work, PCL scaffolds were lithographically fabricated using a sacrificial process to produce submicron-thick scaffolds with openings of specific sizes and shapes (e.g., grid, hexagonal pattern). The HTM cell growth on gelatin-coated PCL scaffolds was assessed by scanning electron microscopy, tetrazolium metabolic activity assay, and cytoskeletal organization of F-actin. Expression of HTM-specific markers and ECM deposition were assessed by immunocytochemistry and qPCR analysis. Gelatin-coated, micropatterned, ultrathin, porous PCL scaffolds with a grid pattern supported proper HTM cell growth, cytoskeleton organization, HTM-marker expression, and ECM deposition, demonstrating the feasibility of using these PCL scaffolds to tissue-engineer implantable, healthy ocular outflow tissue.

## 1. Introduction

Glaucoma is a neurodegenerative disease affecting the optic nerve and is one of the leading causes of irreversible blindness worldwide. In particular, primary open angle glaucoma (POAG), the most common type of glaucoma in the Western world, is associated with malfunctioning trabecular meshwork cells [1]. It is estimated that POAG represents 90% of all glaucoma cases, affecting 57.5 million people in 2015 with an anticipated rise to 111.8 million patients by 2040 [2,3]. As of yet, there is no cure for glaucoma. 

Targeting the pathological site—the human trabecular meshwork (HTM)—represents a promising therapeutic strategy for treatment of glaucoma [4,5,6,7,8], which is the main site of aqueous humor drainage [9]. Aqueous humor is produced by the ciliary processes at an almost constant rate and is eliminated in a pressure-dependent manner through the trabecular meshwork. As a result, resistance to outflow of aqueous humor regulates IOP. Concurrent with damage to the trabecular meshwork cells in glaucoma, their fibrillary extracellular matrix (ECM) is increased [10], leading to the build-up of plaque-like material, which ultimately creates an impediment to fluid outflow from the eye, leading to elevated IOP. The elevated IOP, over time, causes damage to the axons of the retinal ganglion cells, leading to vision loss [9,10]. 

Currently, IOP is the only modifiable risk factor of glaucoma, and reduction of IOP is one of the proven approaches to treating glaucoma. IOP-lowering therapies include laser treatment of the trabecular meshwork, medications, and surgical shunting of the aqueous humor into the downstream outflow pathways or the subconjunctival space [10,11]. A potential alternative to the current strategies for IOP control would be the replacement of the trabecular meshwork with a tissue-engineered HTM cell-scaffold construct that recapitulates the HTM structure and function in vivo. In particular, this approach could serve as a better treatment option since damaged HTM cells and abnormal ECM deposition would be replaced with a healthy biomimetic trabecular meshwork tissue that maintains HTM cell function to regulate the IOP homeostasis. Additionally, this approach can be easily combined with stem cell therapy, which has shown promise in animal models [11,12,13,14,15], by providing a potential delivery vehicle for stem cells, possibly leading to better in vitro stem cell growth and potentially in vivo tissue integration. 

Trabecular meshwork cells in vivo form a complex three dimensional (3D) network of ECM proteins [16,17,18]. It is thought that cells use these networks to provide topographical and mechanical cues, which guide morphological and biological properties of the tissue. In addition, in vivo HTM cells have fluid (i.e., aqueous humor) flowing around them. Thus, it is desirable, and possibly required, to mimic this fluid flow to obtain appropriate gene expression and cellular properties in the in vitro cell culture environment. Once implanted, this tissue-engineered construct will allow the HTM cells to mimic the native cellular structures that they are replacing as closely as possible. A porous membrane or scaffold containing openings allows fluid flow across the cells. Biological and topographical cues affect HTM cell shape and organization [19,20,21], and cell shape and volume are essential to HTM outflow function [22]. However, it is unknown how a specific pore shape of the scaffold affects HTM cell behaviors. 

In order to recreate the in vivo trabecular meshwork, the scaffolding material must be not only biocompatible, biodegradable, and implantable, but also permeable, micropatternable, and able to provide topographical cues to support HTM cell growth and function. We have successfully demonstrated the feasibility of bioengineering the trabecular meshwork in vitro using micropatterned, porous SU-8 scaffolds [23,24,25,26,27]. Theses SU-8 scaffolds provide a valuable platform for recreating in vitro trabecular meshwork models; however, SU-8 is not ideal for in vivo applications because it is not biodegradable. Alternatively, polycaprolactone (PCL) is promising for bioengineering the trabecular meshwork since it is an FDA-approved, biocompatible, biodegradable, and implantable material, having been widely used for cell culture, tissue engineering, cell implantation, and transplantation. 

Porous PCL scaffolds can be fabricated through a variety of methods, including selective laser sintering [28,29], fused deposition modeling [30], salt leaching [31], porogen leaching [32], emission templating [33,34], 3D printing [35,36], gas forming [37], and electrospinning [38]. All these microfabrication methods produce PCL scaffolds either with random pores, or with well-defined pore structures; however, these are too thick or too dense for use in eye tissue engineering. Ultrathin porous membranes are valuable to mimic the basement membrane of delicate ocular structures. We have demonstrated the feasibility of microfabricating 200–500 nm-in-thickness, porous PCL membranes using a sacrificial molding process [39].

Here, a sacrificial layer technique was used to further fabricate micropatterned, ultrathin, porous PCL scaffolds with both grid and hexagonal patterns. In order to promote HTM cell attachment, gelatin was used to coat both scaffolds. We tested the feasibility of recreating trabecular meshwork tissue using these micropatterned PCL scaffolds. Effects of pore structure on HTM cell attachment, growth, morphology, ECM deposition (e.g., fibronectin, collagen IV, laminin), and characteristic marker expression (e.g., α-SMA, myocilin, αB-crystallin) were studied by tetrazolium (MTT) assay, scanning electronic microscopy (SEM), immunocytochemistry, and qPCR analysis. This bioengineered 3D tissue may provide a new avenue for bioengineering the ocular outflow tissue for therapeutic treatment of glaucoma.

## 2. Materials and Methods

### 2.1. Microfabrication of Patterned, Ultrathin, Porous Polycaprolactone (PCL) Scaffolds

PCL scaffolds, approximately 600 nm in thickness, were fabricated using a sacrificial layer technique; the concept of which has been previously described using liftoff resist [39]. Briefly, a patterned silicon dioxide mold was first created, filled with polymer, and then dissolved, leaving behind a patterned polymer structure. A support ring was attached to the scaffold before it was released from the mold to allow for easy handling (Figure 1).

The process was initiated with a 100-mm silicon wafer containing a 1.5–2-µm thick wet-grown silicon dioxide film. A 1.6-µm thick photoresist film (Microposit S1813, Shipley, Rohm & Haas & Dow Chemical Company, Marlborough, MA, USA) was spin-coated onto the wafer as follows: film was spin-coated onto the wafer as follows: ramp to 1500 revolutions per minute (rpm, i.e., 125.8× *g*) at a rate of 3500 rpm (i.e., 684.8× *g*) per second, hold for 5 s ramp to 4000 rpm (i.e., 894.4× *g*), at a rate of 3500 rpm (i.e., 684.8× *g*) per second hold for 40 s. The wafer was then baked for 100 s at 110 °C and exposed at 70 mJ/cm^2^ at 365 nm on a mask aligner using a hard contact. Following the exposure, the wafer was developed for 25 s in metal ion free (MIF) developer (AZ 300MIF, Integrated Micro Materials, Argyle, TX, USA), rinsed with deionized water, and dried. The photoresist thickness was measured using an Alpha-Step contact profilometer (KLA-Tencor, Milpitas California). The wafer was then etched in buffered hydrofluoric acid (BHF) (Transene Company, Danvers, MA, USA) at room temperature to achieve a mold depth of 800–900 nm. The etch rate was approximately 100 nm/min. The photoresist was stripped with acetone and isopropanol, and then the wafer was rinsed with deionized water and dried. The wafer with the completed molds was cleaved and broken into individual dies by hand using a scriber. Rings of SU-8 2010 (MicroChem, Newton, MA, USA) were used to encircle the scaffolds preventing them from folding on themselves and for easy handling with tweezers after they were released. The SU-8 ring was fabricated by coating a silicon wafer with lift-off resist, LOR 3A (MicroChem), at a final spin speed of 750 rpm (i.e., 31.4× *g*) and baking the LOR 3A at 150 °C for 5 min. SU-8 2010 was then coated on top of the LOR 3A at a final spin speed of 750 rpm (i.e., 31.4× *g*) and baked at 65 °C for 1 min and 95 °C for 3 min. The wafer was allowed to sit for 10 min, followed by UV-exposure at 150 mJ/cm^2^ using a transparency mask. The wafer was then baked for 1 min at 65 °C and 10 min at 95 °C, and allowed to cool for 10 min. The wafer was submerged in SU-8 developer, rinsed with isopropanol and deionized water and dried with a dry stream of nitrogen. The wafer was then annealed for 15 min at 150 °C to reflow the resist and to help release any thermal stress. Finally, the completed rings were released in 300MIF developer at room temperature. 

Once the molds and SU-8 rings were fabricated, the silicon dioxide molds were coated with 45 MW PCL (Sigma Aldrich, St. Louis, MO, USA) (0.025 g/mL in toluene) by spin coating using a recipe very similar to that of SC1813 above, with a final spin speed of 1250–2000 rpm (87.3–223.6× *g*) (processing was adjusted to achieve the desired mechanical stability over the course of the experiments). To prevent toluene evaporation during spin coating, two drops of PCL solution was placed on the mold using a plastic pipette and allowed to sit for less than 5 s before the spin process was started. After spin coating, the mold and PCL were baked at 95 °C for at least 2 min to evaporate any residual toluene. During the bake step, an SU-8 2010 ring was placed on the mold and was bonded to the spin-coated PCL by dipping the ring in a 0.2 g/mL PCL in toluene solution before being placed. Finally, the scaffolds were released from the silicon dioxide mold by placing them in BHF overnight, leaving ultrathin, free standing, micropatterned, porous PCL scaffolds.

### 2.2. Human Trabecular Meshwork (HTM) Cell Culture

HTM cells were isolated from donor tissue rings discarded after penetrating keratoplasty. Isolation of the cells was performed under an IRB-exempt protocol approved by the SUNY Downstate Medical IRB (see Institutional Review Board Statement). Before use in experiments, all HTM cells were characterized for expression of αB crystallin and α-smooth muscle actin (α-SMA). HTM cells were initially plated in 75 cm^2^ cell culture flasks with 10% fetal bovine serum (FBS) (Atlas Biologicals, Fort Collins, CO, USA) in Improved MEM (IMEM) (Corning Cellgro, Manassas, VA, USA) supplemented with 1% 10 mg/mL gentamicin. Cell culture medium was changed approximately every 48 h. Cells were maintained at 37 °C in a humidified atmosphere with 5% carbon dioxide until 80–90% confluence, at which point cells were trypsinized using 0.25% trypsin/0.5 mM EDTA (Gibco, Grand Island, NY, USA) and sub-cultured. Primary cell from three separate donors were used in the experiments. All studies were conducted using cells in passages two to five.

### 2.3. Culture of HTM Cells on PCL Scaffolds 

Microfabricated PCL scaffolds were washed with 10% calcium chloride (Sigma Aldrich, St. Louis, MO, USA) to remove any residual fluoride ions from the fabrication process. Scaffolds were then sterilized in 70% ethanol and air dried. Scaffolds designated gelatin-coated were then submerged into a 1% gelatin (Sigma Aldrich, St. Louis, MO, USA) solution in water for 30 min and allowed to air dry. The gelatin coating reduces the hydrophobicity of the scaffolds and promotes cell attachment. The scaffolds were placed in 24-well plates and HTM cells were plated at a cell density of 5 × 10^4^ cells per well. HTM cells on PCL scaffolds were cultured in IMEM (Corning Cellgro, Manassas, VA, USA) containing 10% FBS (Atlas Biologicals, Fort Collins, CO, USA) with 1% 10 mg/mL gentamicin and allowed to grow for up to 14 days. Media were changed approximately every 48 h. 

### 2.4. Scanning Electron Microscopy (SEM)

The structure of the PCL scaffolds, the HTM cell morphology and cell coverage on the PCL scaffolds were characterized using a LEO 1550 field emission SEM (Leo Electron Microscopy Ltd., Cambridge, UK) 7 and 14 days after cultures were initiated. For SEM imaging, samples were fixed with 3% glutaraldehyde solution in 0.1 M phosphate buffer (pH 7.4) containing 0.1 M sucrose for 2 h at room temperature. These samples were then rinsed three times in 0.1 M phosphate buffer, dehydrated in a graded ethanol series and slowly infiltrated with a graded hexamethyldisilazane (HMDS)-ethanol (Sigma Aldrich, St. Louis, MO, USA) series (25%, 50%, 75%, and 100% HMDS) for drying. Fixed samples were mounted on 1-cm^2^ stubs using carbon tape and were sputter-coated with ≥5 nm gold-palladium to avoid charging the sample. Images were collected at a working distance of 3 mm with an acceleration gun voltage of 5 kV. 

### 2.5. MTT Assay

To assess HTM cell growth on PCL scaffolds, an MTT metabolic activity assay (Life Technologies, Eugene, OR, USA) was performed. HTM cells were plated on PCL scaffolds that were fitted in a 24-well plate at cell density of 5 × 10^4^ cells per well and allowed to grow for 3 days. HTM cells grown on glass coverslips and unpatterned PCL discs were used as controls for comparison. On day 3, scaffolds with HTM cells were transferred to new sterile wells and incubated for 4 h with fresh medium supplemented with 12 mM 3-(4,5-Dimethylthiazol-2-yl)-2,5-Diphenyltetrazolium Bromide (MTT dye). An additional 4-h incubation period in 0.01 M HCl with 1% SDS was performed, followed by absorbance measurement at 570 nm using a Tecan Infinite M200 multi-plate reader (Tecan, Männedorf, Switzerland). All experiments were performed in triplicate, using three donor cell strains. 

### 2.6. Optical Imaging

HTM cells grown on PCL scaffolds were imaged on days 2, 5, 10 and 14 using an inverted microscope (Nikon Eclipse TS100, Micro Video Instruments, Avon, MA, USA) equipped with a Retiga 2000R digital camera and QImaging software (QCapture Pro Version 6.0.0.412).

### 2.7. Cytoskeleton Staining and Immunocytochemistry Followed by Confocal Imaging

Cytoskeletal protein F-actin was visualized by phalloidin staining. HTM-specific markers, α-SMA, myocilin, and αB-crystallin, as well as ECM proteins, collagen IV, fibronectin, and laminin were assessed using immunocytochemistry and confocal microscopy. Cells grown on PCL scaffolds for 14 days were fixed with 4% paraformaldehyde, permeabilized with 0.2% Triton X-100 in phosphate buffer saline (PBS) (excluding the samples used to analyze secreted extracellular matrix proteins) and blocked with 5% FBS. These HTM cells were then incubated with phalloidin (1:50) (Life Technologies, Carlsbad, CA, USA) to label F-actin cytoskeleton filaments or the following primary antibodies: rabbit anti-myocilin (1:200), mouse anti-αB-crystallin (1:500), and mouse anti-α-SMA (1:800) (Abcam, Cambridge, MA, USA). ECM proteins were detected using the following primary antibodies: rabbit anti-collagen IV (1:100) (Abcam), mouse anti-fibronectin (1:100) (Sigma Aldrich, St. Louis, MO, USA), and chicken anti-laminin (1:500) (Abcam). Secondary antibodies goat anti-mouse Alexa Fluor^®^ 594, goat anti-rabbit Alexa Fluor^®^ 488, and goat anti-chicken 633 (Invitrogen, Grand Island, NY, USA) were used. All samples were co-stained with 4′,6-diamidino-2-phenylindole (DAPI) (Invitrogen, Grand Island, NY, USA) to detect cell nuclei. Confocal microscopy was performed using a Leica SP5 confocal microscope (Leica Microsystems, Mannheim, Germany), with an oil-immersion objective (64×). Images were processed using Leica LasAF software (Leica Application Suite 2.7.3.9723) and captured using the same settings across all samples. Using ImageJ color threshold analysis, the fluorescence intensity was extracted from confocal images of F-actin, α-SMA, collagen IV, fibronectin, laminin, myocilin and αB-crystalin and DAPI-stained HTM cells grown on PCL scaffolds with grid- and hexagon-shaped pores.

### 2.8. Quantitative Polymerase Chain Reaction (qPCR) Analysis

Total RNA was extracted from samples cultured for 14 days on gelatin-coated coverslips, unpatterned PCL discs, and PCL scaffolds with grid- and hexagon-shaped pores using an RNeasy Plus Mini kit (Qiagen Inc., Valencia, CA, USA). RNA concentrations were determined using a NanoDrop spectrophotometer. 20 ng of RNA per sample was used for each qPCR experiment. qPCR was carried out using a TaqMan RNA-to-CT 1-Step Kit (Applied Biosystems, Carlsbad, CA, USA) and performed on an AB StepOnePlus Real Time PCR system (Life Technologies) using primers for α-SMA, fibronectin, collagen type IV and GAPDH as the housekeeping gene (Table 1). The temperature profile was as follows: 48 °C for 15 min (reverse transcription step), followed by an enzyme activation step of 95 °C for 10 min, 40 cycles of 15 s denaturation at 95 °C, and 1 min of anneal/extend at 60 °C. Relative quantitation data analysis was performed using the comparative quantification method, ΔΔCt, with GAPDH as the endogenous reference. qPCR experiments were performed in triplicate (technical replicates) from duplicate biological experiments for each of the three donor cells. Average values are presented as mean ± SD.

### 2.9. Statistical Analysis

Data were expressed as mean ± standard deviation. The difference between cells cultured on coverslips, PCL unpatterned and patterned scaffolds were analyzed using two-way ANOVA followed by Bonferroni post-hoc testing (GraphPad Prism 6.02; GraphPad Software, Inc., La Jolla, CA, USA). *p* < 0.05 was considered significant. 

## 3. Results

### 3.1. Micropatterned, Ultrathin, Porous PCL Scaffolds Support Better Trabecular Meshwork Cell Growth Than 2D Glass Coverslips

Patterned topography has been previously shown to affect the orientation and alignment of HTM cell layers, as well as the morphology of these cells [20]. To assess whether cell growth was affected by the scaffold’s micropatterned pore shape, porous PCL scaffolds with grid and hexagonal patterns were fabricated (Figure 2a,b). The pore size of the grid and hexagonal patterns were 12 µm (Figure 2c,d). Both PCL grid- and hexagon-patterned scaffolds were 200–300 nm in thickness (Figure 2e,f). These scaffolds were designed for a dual purpose, to provide a topography that better simulates the conditions of the in vivo trabecular meshwork, as well as to provide a porous support structure for cell growth and deposition of the ECM network, allowing for effective fluid flow. Notably, the fluid flow around HTM cells relies mainly on cellular behavior and microenvironment surrounding the cells (e.g., secreted ECM). We found that HTM viable cell density measured by MTT assay was enhanced after culturing on PCL scaffolds, either unpatterned or patterned, when compared to HTM cells cultured on coverslips for three days (Figure 3). Interestingly, cells grown on patterned porous PCL scaffolds demonstrated greater viable cell density compared to those grown on unpatterned PCL scaffolds (*p* < 0.0001, ANOVA, *p* < 0.001 and *p* < 0.01, Bonferoni for the grid and hexagonal pattern, respectively) as shown in Figure 3. These data indicated that micropatterned PCL scaffolds support better cell growth than conventional 2D cultures on glass coverslips or unpatterned PCL discs. This enhanced HTM growth may be due to enhanced mass transfer and/or improved spatial cellular organization on porous PCL scaffolds. The organized pores that make up the scaffolds may provide appropriate channels for movement of signaling molecules, nutrients, and metabolic waste as well as enhanced cell-cell and cell-matrix interactions that increase cell attachment and growth. 

### 3.2. Gelatin-Coated, Micropatterned PCL Scaffolds Support Trabecular Meshwork Cell Morphology and Cytoskeletal Protein Expression

Because of the natural hydrophobicity of PCL scaffolds, there have been a plethora of methods developed to modify their interactions with cells and tissue. We chose gelatin (i.e., denatured collagen) to coat grid- or hexagon-patterned PCL scaffolds, which is one of the most popular surface modifications of PCL and is known to increase cell attachment, hydrophilicity, water uptake, and mechanical strength [29,40,41].

HTM cells were cultured on gelatin-coated, porous PCL scaffolds with grid and hexagonal patterns, respectively, for 14 days. HTM cell growth on these micropatterned PCL scaffolds was monitored by optical imaging at 2, 5, 10 and 14 days, showing that HTM cells grew into a continuous layer of spindle-shaped cells on porous PCL scaffolds of both patterns. Although initial light microscopy showed no apparent differences between gelatin-coated grid- and hexagonal-patterned PCL scaffolds with respect to the attachment or growth of HTM cells (Figure 4), SEM analysis demonstrated dramatic differences in the HTM cell layer morphology between cells grown on the two different patterns (Figure 5a,b). PCL scaffolds fabricated with the grid pattern supported dense HTM cell layer formation compared to the hexagonal pattern (Figure 5b), highlighting the importance of using multiple imaging techniques for studying HTM cell coverage on scaffolds. 

Gelatin coating of micropatterned PCL scaffolds further enhanced cell attachment and growth (Figure 5a) when compared to uncoated patterned PCL scaffolds (Figure 5b). In-depth SEM analysis of HTM cells grown on gelatin-coated grid-patterned and hexagonal-patterned PCL scaffolds demonstrated that the grid pattern promotes better cell coverage over the entire scaffold (Figure 5a, top panel) than the hexagonal pattern (Figure 5a, bottom panel). 

The effect of gelatin coatings on scaffolds was further evaluated in terms of expression of cytoskeletal proteins, F-actin and α-SMA. The actin family of proteins plays a large role in modulating the outflow of aqueous humor. One protein in particular, α-SMA, has a large role in the regulation of cell contractility and in affecting the expression of ECM proteins. α-SMA, expressed primarily in fibroblasts that are responsible for the secretion of ECM [18,20], is also expressed in HTM cells [21]. F-actin of the HTM plays an important role in cytoskeletal organization and aqueous humor outflow regulation. The effect of the pore pattern on F-actin and α-SMA was analyzed. HTM cells grown on gelatin coated glass coverslips were used as control (Appendix A). Results demonstrated that HTM cells grown on grid-patterned porous PCL scaffolds consistently express greater F-actin (16 ± 0.8 vs. 11.7 ± 1.2, *N* ≥ 8) and α-SMA (4.8 ± 10.6 vs. 1.5 ± 0.4, *N* ≥ 8) than those grown on hexagon-patterned PCL scaffolds (Figure 5d and Appendix A). Gelatin coating of porous PCL scaffolds further enhanced phalloidin staining of F-actin, with HTM cells grown on gelatin-coated grid-patterned scaffolds showing more intense staining than those grown on hexagon-patterned gelatin-coated PCL scaffolds (19.2 ± 0.6 vs. 10.5 ± 0.4) (Figure 5c and Appendix A). The confocal images showed diffuse expression of α-SMA across the cell cytoplasm in HTM cells cultured on porous PCL scaffolds with a grid pattern. Interestingly, on gelatin-coated hexagonal patterned PCL scaffolds, α-SMA expression was lower than that of gelatin-coated grid-patterned porous PCL scaffolds (6 ± 0.4 vs. 1.4 ± 0.3), and this protein appeared to be arranged into fibrous structures that colocalized with F-actin fibers (Figure 5c). Since cytoskeleton organization and remodeling is very important for IOP regulation, we chose gelatin-coated scaffolds for the subsequent studies.

### 3.3. Gelatin-Coated PCL Scaffolds Support Higher Gene Expression of ECM Molecules and α-SMA in HTM Cells Than 2D Glass Coverslips

Next, we studied gene expression of the representative HTM-marker, α-SMA, and ECM molecules (fibronectin and collagen IV) for HTM cells cultured on gelatin-coated, micropatterned PCL scaffolds. α-SMA, a mechanosensitive protein that is recruited to stress fibers under high tension [20], is expressed in cells throughout the trabecular meshwork [42]. TM cells are also known to secrete ECM proteins including fibronectin, collagen IV, and laminin [43,44]. 

α-SMA, fibronectin, and collagen IV gene expression were compared between gelatin-coated unpatterned PCL scaffolds and glass coverslips (Figure 6a) and between micropatterned PCL scaffolds (grid and hexagonal pore patterns) and glass coverslips (Figure 6b). HTM cells grown on gelatin-coated unpatterned PCL scaffolds exhibited significantly higher gene expression of α-SMA, fibronectin, and collagen IV at mRNA level than on glass coverslip controls (Figure 6a). As shown in Figure 6b, α-SMA expression was higher in grid-patterned porous PCL scaffolds versus glass coverslips, but there was no significant difference between the hexagonal PCL scaffold and glass (*p* < 0.001 and *p* > 0.05, ANOVA Bonferroni post-test, respectively). The expression of fibronectin also only increased on grid-patterned porous PCL scaffolds when compared to glass (*p* < 0.05, ANOVA Bonferroni post-test). Collagen IV expression increased for both PCL scaffold patterns versus glass (*p* < 0.001 for grid pattern and *p* < 0.0005 for hexagon pattern, ANOVA Bonferroni post-test). Overall, HTM cells cultured on PCL scaffolds showed greater α-SMA, fibronectin, and collagen IV expression when compared to cells grown on glass coverslips (*P* < 0.001 for all, ANOVA Bonferroni post-test), thereby reinforcing PCL superiority over glass as scaffold material for HTM cells. 

### 3.4. Gelatin-Coated Grid-Patterned PCL Scaffolds Support Higher Expression of ECM Proteins and HTM Markers Than Hexagon-Patterned Scaffolds

We further evaluated effects of micropatterned pore structures on characteristic ECM deposition (collagen IV, fibronectin, and laminin) and HTM marker (myocilin and αB-crystalin) expression using confocal microscopy of immunostained HTM cells grown on gelatin-coated, micropatterned PCL scaffolds (Figure 7). The expression of ECM proteins can largely determine the ability of the trabecular meshwork to modulate the flow of aqueous humor across the HTM and the inner wall of the Schlemm’s canal [45,46]. Trabecular meshwork cells are known for their ability to prolifically produce and remodel their ECM. HTM cells grown on porous PCL scaffolds with grid pattern expressed fibrous collagen IV, fibronectin, and laminin across the entire scaffold compared to those grown on hexagonal pore pattern, where fibronectin and laminin proteins were fewer and appear diffuse (Figure 7a).

On the grid-pore patterned scaffolds, laminin protein colocalized beneath the fibronectin fibers in certain areas, whereas this colocalization was not seen on the PCL scaffolds with the hexagonal pattern. Additionally, on the porous PCL scaffolds with the grid pattern, HTM cells were embedded within the ECM material secreted by the cells, forming 2–3 cell layers in a 3-dimensional structure comprised of HTM cells and their surrounding ECM network (Figure 7b), while on the hexagon-patterned PCL scaffolds, HTM cells grew as a monolayer and cells were surrounded by less ECM (Figure 7c).

In addition to α-SMA (see Figure 4), the expression profiles of HTM marker proteins, myocilin and αB-crystallin, were also evaluated (Figure 8). Myocilin was expressed by HTM cells cultured on both micropatterned porous PCL scaffolds with grid and hexagonal pattern. HTM cells grown on PCL scaffolds with the grid pattern exhibited more intense, cytoplasmic myocilin expression than those grown on hexagon-patterned scaffolds. Although not as prominent as myocilin, αB-crystallin was expressed in cells grown on the grid-patterned porous scaffolds (5.2 ± 0.6, *N* ≥ 10), but barely detectable in cells grown on the hexagonal pore-patterned scaffolds (1.1 ± 0.4, *N* ≥ 10). This difference may be due to the differences in cell coverage between grid and hexagonal pores as described above.

## 4. Discussion

In this study, it is clearly demonstrated that gelatin-coated, micropatterned, ultrathin, porous PCL scaffolds with grid patterns support appropriate HTM cell growth, cytoskeleton organization, ECM deposition, and HTM-marker expression across the scaffold, simulating the 3D structure of the trabecular meshwork. The grid pore structure of micropatterned PCL scaffolds provides the topographical cues required for HTM cells to properly express characteristic HTM markers, such as α-SMA, myocilin, and αB-crystallin, and ECM proteins, such as collagen IV, fibronectin, and laminin. Expression of HTM-specific markers myocilin, αB-crystallin, as well as α-SMA are important to both the function and characterization of trabecular meshwork cells [47,48,49]. Myocilin is one of the most abundant proteins expressed in the TM [25], and mutations in the structure of this protein have been shown to deregulate IOP [42,50,51]. The protein αB-crystallin is expressed in most ocular tissues. αB-crystallin’s primary function is as a heat-shock and chaperone protein [49]; this protein is expressed in TM cells and can be induced by mechanical stretch [52]. Trabecular meshwork cells are also known for their ability to prolifically produce and remodel their ECM. These porous PCL scaffolds with grid pattern thus appear to support the potential creation of a 3D HTM with properties that are comparable to HTM cells grown on our previously developed SU-8 scaffolds.

Our previous studies have demonstrated that gelatin-coated micropatterned photoresist SU-8 scaffolds with grid (12-µm in pore size) supported HTM cells to recapitulate cell phenotype and outflow physiology of the trabecular outflow tissue [23,24,25,26,27,52]. For example, HTM cells on these SU-8 scaffolds grow into multi-layer structures, express characteristic HTM markers (e.g., α-SMA, myocilin, αB-crystallin) and deposit ECM proteins (e.g., collagen IV, fibronectin) on scaffolds, and exhibit simulated outflow facility comparable to human eye level [23]. These cells grown on SU-8 scaffolds are responsive to pharmacological agents, such as IOP-elevating agents (e.g., dexamethasone, TGFβ) and IOP-lowering agents (e.g., ROCK inhibitor Y27632) [25,52]. It has provided a functional in vitro model of the conventional outflow pathway that allows monitoring of both cell phenotype and simulated outflow facility [24,53]. The relevance of this in vitro HTM-based model has been further confirmed by utilizing it as an essential complementary approach to animal studies of potential outflow regulators (e.g., TRPV4 [54], nitric oxide [55]) in which in vitro outflow results were confirmed with in vivo animal studies. Although micropatterned SU-8 scaffolds provide an excellent in vitro HTM model system, SU-8 is non-biodegradable and not ideal for implantation applications.

Micropatterned PCL scaffolds provide an alternative biodegradable, implantable option to trabecular meshwork tissue engineering. The ocular biocompatibility and in vivo safety of porous PCL films or scaffolds have been well documented as ophthalmic implants [56,57,58,59,60,61]. In order to make our implantable HTM cell-scaffold construct a viable approach to trabecular outflow tissue regeneration, in vivo studies of its biocompatibility and ability to control the IOP need to be done in the future. Furthermore, more in vitro studies should be done to test whether HTM cells remain on the support material or detach with simulated aqueous humor flow. We have found that HTM cells grown on SU-8 scaffolds remained as a confluent cell layer on the gelatin-coated, micropatterned SU-8 scaffolds and maintained HTM cell morphology and cytoskeleton organization after perfusion at 40 mL/min in the apical-to-basal direction for 4 h (corresponding to transmembrane pressure of 9.0 ± 1.0 mmHg) [23]. Based on this previous study, we predict that HTM cells grown on the gelatin-coated, micropatterned PCL scaffold would remain on it without detachment under simulated aqueous humor flow. This prediction can be confirmed by both in vitro and in vivo experiments. Additionally, the aqueous humor in glaucoma patients is highly enriched with free radicals and induces intense oxidative stress in the anterior segment of the eye. Therefore, it needs to be tested whether even under these simulated “toxic” conditions the HTM cells still remain on the PCL scaffold. Altogether, it will confirm the utility of implantable, micropatterned, ultrathin, porous PCL scaffolds for HTM cells to develop into an outflow tissue for trabecular meshwork regeneration.

## 5. Conclusions

By surveying the morphological and biological properties of HTM cells, we were able to show the feasibility of using micropatterned, ultrathin, porous PCL as a viable scaffold for properly culturing HTM cells. The scaffold pore-structure shape has considerable effects on the cell growth, F-actin cytoskeleton organization, HTM marker expression, and ECM deposition. Additionally, the use of gelatin as scaffold coating enhanced cell attachment and cell proliferation. This work represents the initial steps towards developing an implantable, biomimetic healthy HTM that could serve as novel implant to maintain patient trabecular meshwork outflow function.

## Figures and Tables

**Figure 1 bioengineering-10-00679-f001:**
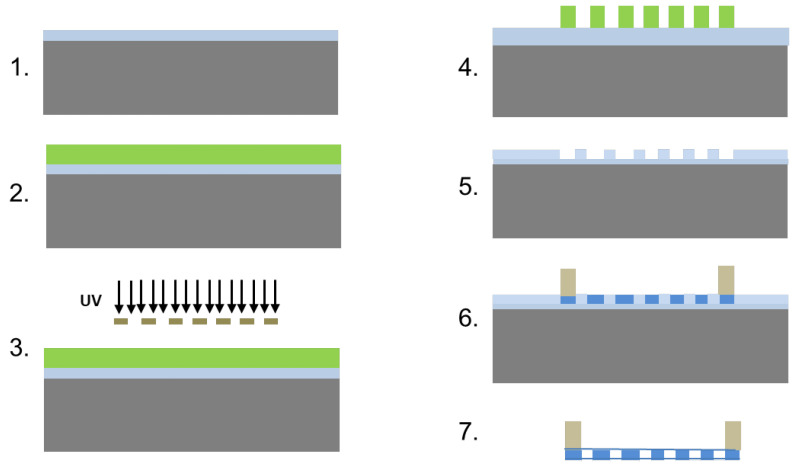
Schematic overview of the scaffold fabrication. Step 1. Start with a silicon wafer with a 1–2 µm thick silicon dioxide layer. 2. Spin coat SC1813 on wafer surface and soft bake. 3. Expose the SC1813 through chrome mask with desired patterns. 4. Develop. 5. Etch the silicon dioxide with buffered hydrofluoric acid and strip the SC1813 from the wafer surface using acetone. 6. Spin coat PCL solution on the etched mold to create through holes and attach SU8 ring to edges of the PCL scaffold by dipping the ring in concentrated PCL for bonding with the scaffold. 7. Release the scaffold by etching overnight in buffered HF. The SU-8 ring keeps the scaffold from folding and allows for handling.

**Figure 2 bioengineering-10-00679-f002:**
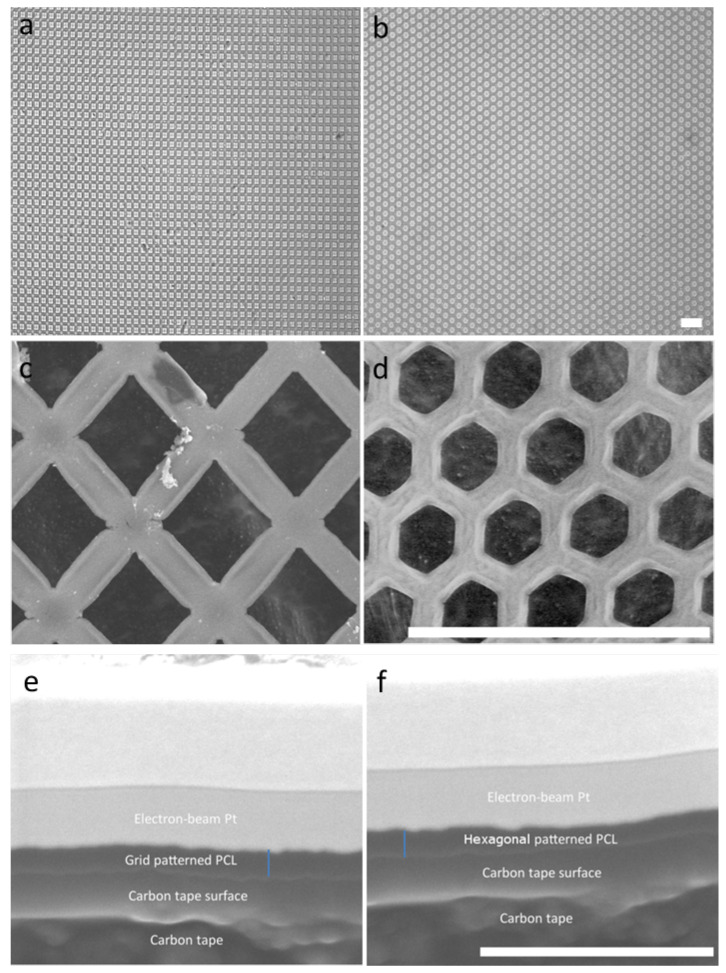
Micropatterned porous PCL scaffolds. Optical images of the PCL scaffold with the grid (**a**) or hexagonal (**b**) pattern. Scanning electron micrographs of grid- (**c**) and hexagon-patterned (**d**) scaffolds. Scale bar = 40 µm. The cross section of grid- (**e**) and hexagon-patterned (**f**) scaffolds. Blue line: measurement of scaffold thickness. Scale bar = 2 µm.

**Figure 3 bioengineering-10-00679-f003:**
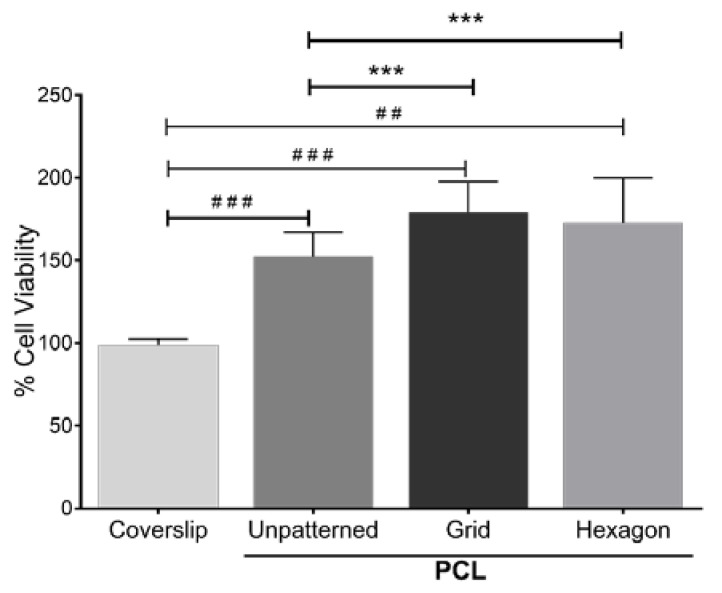
MTT assay of HTM cell growth on micropatterned porous PCL scaffolds for 3 days. HTM cells grown on glass coverslips or unpatterned PCL discs were used as controls. ##, *p* < 0.01. ###, *p* < 0.001. ***, *p* < 0.001.

**Figure 4 bioengineering-10-00679-f004:**
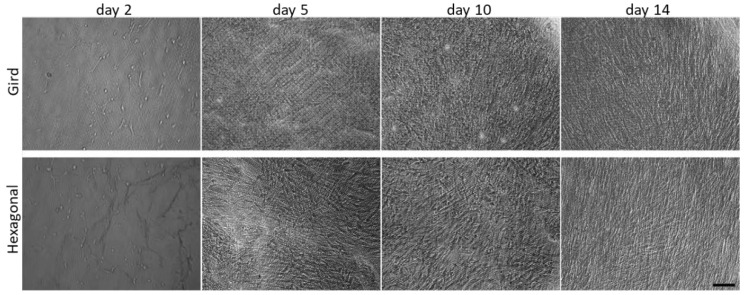
Optical images showing time course of HTM cells cultured on micropatterned porous PCL scaffolds up to 14 days. Scale bar = 100 µm.

**Figure 5 bioengineering-10-00679-f005:**
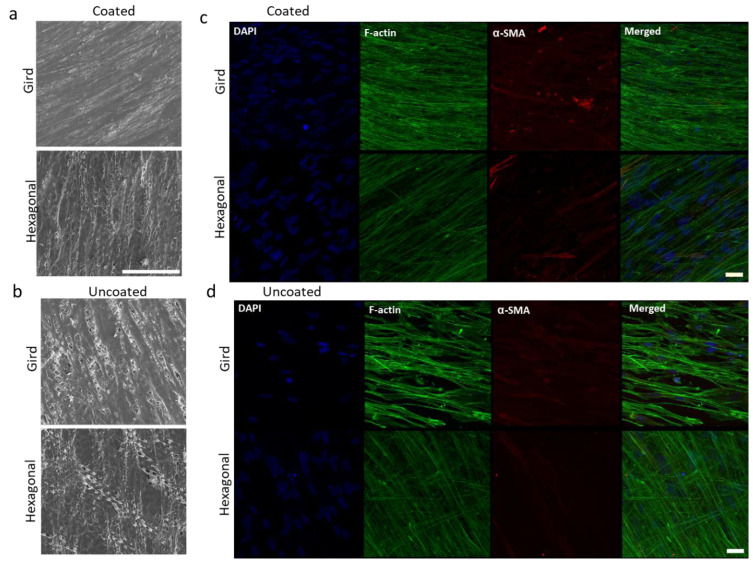
Effects of gelatin coating on HTM cells grown on micropatterned porous PCL scaffolds for 14 days. (**a**,**b**) SEM images of HTM cells grown on PCL scaffolds. Scale bar = 100 µm. (**c**,**d**) Confocal imaging of F-actin (green) and α-SMA (red) expression of HTM cells grown on PCL scaffolds. Scale bar = 30 µm. (**a**,**c**) Coated with gelatin. (**b**,**d**) Uncoated controls. Blue: Nuclei.

**Figure 6 bioengineering-10-00679-f006:**
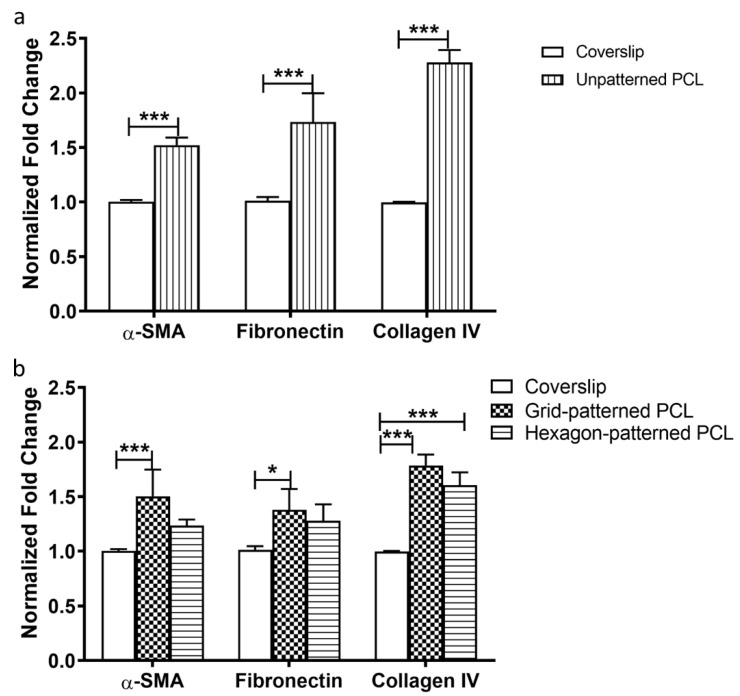
qPCR analysis of gene expression of α-SMA, fibronectin and collagen IV. (**a**) HTM cells cultured on unpatterned PCL normalized to those on glass coverslips. (**b**) HTM cells cultured on micropatterned porous PCL scaffolds normalized to those on glass coverslips. *, *p* < 0.05. ***, *p* < 0.001.

**Figure 7 bioengineering-10-00679-f007:**
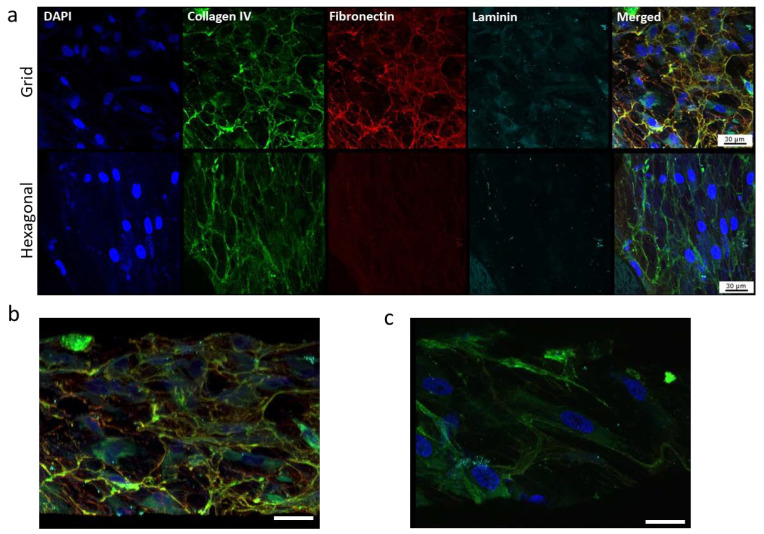
Confocal images showing the effect of pore structures on ECM deposition. (**a**) Expression of ECM proteins, collagen, fibronectin and laminin of HTM cells grown on gelatin-coated, micropatterned, porous PCL scaffolds. (**b**,**c**) Tilted angle view of 3D reconstruction of HTM cells grown on the grid-patterned PCL scaffold (**b**) and hexagon-patterned PCL scaffold (**c**). Green, collagen IV; red, fibronectin; cyan, laminin; blue, DAPI-stained nuclei. Scale bar = 30 µm.

**Figure 8 bioengineering-10-00679-f008:**
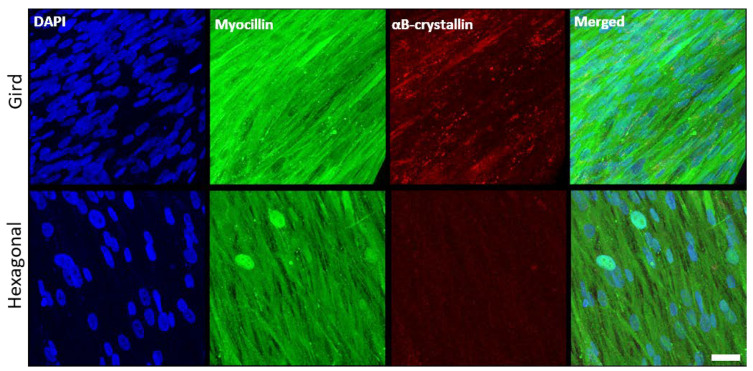
Confocal images showing the effect of pore structures on marker expression of HTM cells grown on gelatin-coated, micropatterned, porous PCL scaffolds. (**Top Panel**) Grid-patterned PCL scaffolds. (**Bottom Panel**) Hexagon-patterned PCL scaffolds. Blue, DAPI-stained nuclei; green, myocilin; red, αB-crystallin. Scale bar = 30 µm.

**Table 1 bioengineering-10-00679-t001:** Primer sequences for qPCR.

Gene	Forward	Reverse
α-SMA	5′-GGA TTA AGT TCA TAA GAT TCC ATG CT-3′	5′-TGT TAT GGA GAG TGG CAG AAA G-3′
Fibronectin	5′-GTC CTT GTG TCC TGA TCG TTG-3′	5′-AGG CTG GAT GAT GGT AGA TTG-3′
Collagen IV	5′-CCT TTG TGC CAT TGC ATC C-3′	5′-GAA CAA AAG GGA CAA GAG GAC-3′
GAPDH	5′-TGT AGT TGA GGT CAA TGA AGG D-3′	5′-ACA TCG CTC AGA CAC CAT G-3′

## Data Availability

The data presented in this study are available in the article.

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
