# Peer review of "Recreating the Trabecular Outflow Tissue on Implantable, Micropatterned, Ultrathin, Porous Polycaprolactone Scaffolds"

_bioengineering, 2023, doi:10.3390/bioengineering10060679_

Round 1

Reviewer 1 Report

The authors describe the development of a PCL-fabricated scaffold and have described the effects of altering the topography of the scaffold on the attachment and viability of the HTMs. Overall the experimental rationale is logical and the data well presented but further experiments with control data needs to be included.      

Materials and methods:

The methods section is well written with full experimental details provided.

Where rpm is cited, needs to be reported as ‘x g’

Use of human cells is described from human tissues, the ethical approval / consent process is not very explicit. Needs to have its own section.

Results:

Coverslip control data is missing from data presented in figure 7. Similarly in figure 8 immunohistochemistry has been performed on PCL / grid scaffolds, but there is no comparison for the PCL/ hexagonal and coverslips. This data needs to be included.

Author Response

Reviewer 1

The authors describe the development of a PCL-fabricated scaffold and have described the effects of altering the topography of the scaffold on the attachment and viability of the HTMs. Overall the experimental rationale is logical and the data well presented but further experiments with control data needs to be included.      

Materials and methods:

The methods section is well written with full experimental details provided.

Where rpm is cited, needs to be reported as ‘x g’

Response: The rpm was also reported as x g.

Use of human cells is described from human tissues, the ethical approval / consent process is not very explicit. Needs to have its own section.

Response: An Institutional Review Board Statement was revised, making it read as follows.

Institutional Review Board Statement: The protocol for isolation of HTM cells from donor tissue rings discarded after penetrating keratoplasty was IRB-exempt and approved by the SUNY Downstate Medical IRB (Approval number: ).

Informed Consent Statement: Not applicable since it was IRB-exempt.

Results:

Coverslip control data is missing from data presented in figure 7. Similarly in figure 8 immunohistochemistry has been performed on PCL / grid scaffolds, but there is no comparison for the PCL/ hexagonal and coverslips. This data needs to be included.

Response: The coverslip control data for Figure 7 were included in Supplementary Materials (Figure S1) and titled view of reconstructed confocal image showing expression of ECM proteins from HTM cells cultured on the gelatin-coated, porous PCL scaffold with the hexagonal pattern was included as Figure 7c while Figure 8 was changed as Figure 7b.

Reviewer 2 Report

The present work is of great interest because it shows a new way in glaucoma surgery and has a very physiological starting point. Especially with such an active tissue as the trabecular meshwork, this approach, possibly with stem cells, is a great idea.

However, I have the following comments on the paper:

1. line 47-48: This statement is not correct, because in the meantime other therapies, like citiculine and coenzyme Q10, can successfully slow down glaucoma progression. In this respect, pressure reduction is one possible form of therapy, but not the only one.

2. The results with HTM cells unter resting conditions are very promising. However, in the discussion I would still state 2 limitations:

a. under an artificial eye model, it would have to be tested whether even with simulated aqueous humor flow, the HTM cells remain on the support material or detach. 

b. the aqueous humor in glaucoma patients is highly enriched with free radicals and induces intense oxidative stress in the anterior segment on the eye. It would have to be tested whether even under these simulated "toxic" conditions the HTM cells still remain on the carrier material.

Author Response

Reviewer 2

The present work is of great interest because it shows a new way in glaucoma surgery and has a very physiological starting point. Especially with such an active tissue as the trabecular meshwork, this approach, possibly with stem cells, is a great idea.

However, I have the following comments on the paper:

1. line 47-48: This statement is not correct, because in the meantime other therapies, like citiculine and coenzyme Q10, can successfully slow down glaucoma progression. In this respect, pressure reduction is one possible form of therapy, but not the only one.

Response: The statement was revised making it read, “Currently, IOP is the only modifiable risk factor of glaucoma, and reduction of IOP is one of the proven approaches to treating glaucoma.”

2. The results with HTM cells under resting conditions are very promising. However, in the discussion I would still state 2 limitations:

a. under an artificial eye model, it would have to be tested whether even with simulated aqueous humor flow, the HTM cells remain on the support material or detach. 

b. the aqueous humor in glaucoma patients is highly enriched with free radicals and induces intense oxidative stress in the anterior segment on the eye. It would have to be tested whether even under these simulated "toxic" conditions the HTM cells still remain on the carrier material.

Response: We thank the reviewer for point out these limitations. We included these two points in Discussion.

Reviewer 3 Report

This manuscript demonstrate the technique to recreate the trabecular outflow tissue on scaffolds, some questions had been raised to be clarified:

1. The introduction is lengthy and needs to be simplified. Authors should emphasize the importance of HTM and the advantage of micropatterned PCL scaffolds.

2. The low picture quality and the lack of quantification in the data fail to support the author's conclusions.

3. The current experiments and data presented are inadequate.

4. Authors should write the discussion of the results in a separate chapter. The current discussion is disordered and insufficient

5. The structure of the manuscript is illogical. For example, the meaning of 3.1 "Trabecular Meshwork Cell Growth on Micropatterned, Ultrathin, Porous PCL Scaffolds" and 3.3 "Effects of Micropatterned Pore Structure on HTM Cells Grown on PCL Scaffolds" is similar. Meanwhile, the order of the figures is not logically arranged according to the manuscript.

Author Response

Reviewer 3

This manuscript demonstrate the technique to recreate the trabecular outflow tissue on scaffolds, some questions had been raised to be clarified:

  1. The introduction is lengthy and needs to be simplified. Authors should emphasize the importance of HTM and the advantage of micropatterned PCL scaffolds.

Response: The introduction was shortened and streamlined for clarity.

  1. The low picture quality and the lack of quantification in the data fail to support the author's conclusions.

Response: The confocal images of expression of F-actin, α-SMA, ECM proteins (collagen IV, fibronectin, and laminin), and HTM markers (myocilin and αB-crystalin) were quantified by ImageJ analysis and included in Supplementary Materials (Figure S2)

  1. The current experiments and data presented are inadequate.

Response: They were revised to include new data as suggested by reviewer 1 and quantified data as suggested by reviewers 1 and 3.

  1. Authors should write the discussion of the results in a separate chapter. The current discussion is disordered and insufficient

Response: The discussion was separated from the Results section and rewrote.

  1. The structure of the manuscript is illogical. For example, the meaning of 3.1 "Trabecular Meshwork Cell Growth on Micropatterned, Ultrathin, Porous PCL Scaffolds" and 3.3 "Effects of Micropatterned Pore Structure on HTM Cells Grown on PCL Scaffolds" is similar. Meanwhile, the order of the figures is not logically arranged according to the manuscript.

Response: We have reorganized the result section, changed subtitles, and re-arranged the order of figures, for example, switching Figure 4 and 5, separating Figure 7b from Figure 7 and making it as new Figure 8, and changing old Figure 8 to Figure 7b.

Reviewer 4 Report

This is a well-designed study and nicely written manuscript  to fabricate micropatterned, ultrathin, porous PCL scaffolds with both grid and hexagonal patterns, coated with gelatin. Authors tested the feasibility of recreating trabecular meshwork tissue using these micropatterned PCL scaffolds by scanning EM, qPCR and IHC stainings. 

Before publication, several questions should be addressed.

1. This is an in vitro based study, is there any in vivo study to demonstrate the safety of this transplant scaffolds in animal model?

2. In P8L302,  The effect of the pore pattern on F-actin and α-SMA was analyzed, demonstrating that HTM cells  grown on grid-patterned porous PCL scaffolds consistently expressed greater F-actin and α-SMA than those grown on hexagonal-patterned PCL scaffolds (Figures 4c and 4d). We can not be convinced just by the representatives of figure 4, authors should make a quantitative comparison.

3. P13L393,  αB-crystallin was expressed in cells grown on the grid-patterned porous scaffolds but not in cells grown on the hexagonal  pore-patterned scaffolds. The figure7b is hard to be differentiated, authors may provide more evidence such as quantitative data or better quality figure

Author Response

Reviewer 4

This is a well-designed study and nicely written manuscript  to fabricate micropatterned, ultrathin, porous PCL scaffolds with both grid and hexagonal patterns, coated with gelatin. Authors tested the feasibility of recreating trabecular meshwork tissue using these micropatterned PCL scaffolds by scanning EM, qPCR and IHC stainings. 

Before publication, several questions should be addressed.

  1. This is an in vitro based study, is there any in vivo study to demonstrate the safety of this transplant scaffolds in animal model?

Response: We did not perform in vivo study using these micropatterned PCL scaffolds. There were literatures to demonstrate the safety of PCL scaffolds in animal models. Therefore, in this revision we cited representative papers using PCL porous films or scaffolds for in vivo study to discuss the potential of these micropatterned PCL scaffolds.

  1. In P8L302,  The effect of the pore pattern on F-actin and α-SMA was analyzed, demonstrating that HTM cells  grown on grid-patterned porous PCL scaffolds consistently expressed greater F-actin and α-SMA than those grown on hexagonal-patterned PCL scaffolds (Figures 4c and 4d). We can not be convinced just by the representatives of figure 4, authors should make a quantitative comparison.

Response: Expression of F-actin and α-SMA was quantified by ImageJ analysis of confocal images of immunostained cells on grid- and hexagonal-patterned PCL scaffolds. We revised the sentence as follows.

“The effect of the pore pattern on F-actin and α-SMA was analyzed, demonstrating that HTM cells grown on grid-patterned porous PCL scaffolds consistently expressed greater F-actin (16 ± 0.8 vs. 11.7 ± 1.2, N≥12) and α-SMA (4.8 ± 10.6 vs. 1.5 ± 0.4, N≥12)than those grown on hexagonal-patterned PCL scaffolds (Figures 5c and 5d).”

  1. P13L393,  αB-crystallin was expressed in cells grown on the grid-patterned porous scaffolds but not in cells grown on the hexagonal  pore-patterned scaffolds. The figure7b is hard to be differentiated, authors may provide more evidence such as quantitative data or better quality figure

Response: Expression of ECM proteins and HTM markers (myocilin and αB-crystallin) was quantified by ImageJ as well.

“αB-crystallin was expressed in cells grown on the grid-patterned porous scaffolds (5.2 ± 0.6, N≥10) but barely detectable in cells grown on the hexagonal pore-patterned scaffolds (1.1 ± 0.4, N≥10).”

Round 2

Reviewer 1 Report

corrects appear to have been carried out

Reviewer 4 Report

No further comments